# Mitochondrial Dysfunction as a Driver of Cognitive Impairment in Alzheimer’s Disease

**DOI:** 10.3390/ijms22094850

**Published:** 2021-05-03

**Authors:** Chanchal Sharma, Sehwan Kim, Youngpyo Nam, Un Ju Jung, Sang Ryong Kim

**Affiliations:** 1School of Life Sciences, Kyungpook National University, Daegu 41566, Korea; chanchalmrt@gmail.com; 2BK21 FOUR KNU Creative BioResearch Group, Kyungpook National University, Daegu 41566, Korea; 3Brain Science and Engineering Institute, Kyungpook National University, Daegu 41404, Korea; arputa@naver.com (S.K.); blackpyo2@naver.com (Y.N.); 4Department of Food Science and Nutrition, Pukyong National University, Busan 48513, Korea; jungunju@pknu.ac.kr

**Keywords:** Alzheimer’s disease, mitochondria, free radical, mitophagy, calcium buffering

## Abstract

Alzheimer’s disease (AD) is the most frequent cause of age-related neurodegeneration and cognitive impairment, and there are currently no broadly effective therapies. The underlying pathogenesis is complex, but a growing body of evidence implicates mitochondrial dysfunction as a common pathomechanism involved in many of the hallmark features of the AD brain, such as formation of amyloid-beta (Aβ) aggregates (amyloid plaques), neurofibrillary tangles, cholinergic system dysfunction, impaired synaptic transmission and plasticity, oxidative stress, and neuroinflammation, that lead to neurodegeneration and cognitive dysfunction. Indeed, mitochondrial dysfunction concomitant with progressive accumulation of mitochondrial Aβ is an early event in AD pathogenesis. Healthy mitochondria are critical for providing sufficient energy to maintain endogenous neuroprotective and reparative mechanisms, while disturbances in mitochondrial function, motility, fission, and fusion lead to neuronal malfunction and degeneration associated with excess free radical production and reduced intracellular calcium buffering. In addition, mitochondrial dysfunction can contribute to amyloid-β precursor protein (APP) expression and misprocessing to produce pathogenic fragments (e.g., Aβ1-40). Given this background, we present an overview of the importance of mitochondria for maintenance of neuronal function and how mitochondrial dysfunction acts as a driver of cognitive impairment in AD. Additionally, we provide a brief summary of possible treatments targeting mitochondrial dysfunction as therapeutic approaches for AD.

## 1. Introduction

Despite decades of research, our knowledge of Alzheimer’s disease (AD) pathogenesis remains incomplete. Hardy and Allsop, 1991, proposed the most accepted hypothesis that amyloid-β precursor protein (APP) misprocessing and deposition of pathogenic Aβ fragments is the primary driver of the progressive neuron and synapse loss underlying cognitive decline [1]. However, this hypothesis cannot fully explain disease epidemiology and clinical course. For instance, amyloid accumulation may occur in brains with no clinical manifestations, and elevated Aβ levels are not always consistent with clinical severity [2,3,4], suggesting that Aβ deposition may not be specific to AD [5,6]. Further, AD animal model studies have failed to establish that Aβ fibril accumulation is necessary and sufficient for disease progression and neuronal death [7]. Rather, Aβ oligomers may be critical cytotoxic agents contributing to AD development together with other pathomechanisms [8]. For instance, a study of Aβ42-overexpressing BRI2-Aβ mice found no impairment in cognitive function or neurodegeneration despite the presence of Aβ oligomers and Aβ amyloid fibrils [9,10]. Furthermore, anti-amyloid drugs have demonstrated no significant benefits in clinical trials [11]. Thus, whether amyloidosis is the primary cause or secondary to other associated pathological processes is still debated. These findings suggest additional mechanisms, potentially involving tau hyperphosphorylation, dysfunction of cholinergic pathways, neuroinflammation, oxidative stress, synaptic failure, and mitochondrial dysfunction [12,13,14,15]. Among these, mitochondrial dysfunction is implicated in a myriad of pathogenic cellular processes, including reactive oxygen species (ROS) generation and ensuing oxidative stress, intracellular calcium deregulation, and apoptosis, strongly suggesting involvement either as a precipitating factor or driver of AD progression [16,17,18,19]. These associations with mitochondrial dysfunction stem from the necessity for sufficient production of adenosine triphosphate (ATP) and biosynthetic intermediates to sustain antioxidant capacity, calcium buffering, and various reparative processes, as well as the direct link between mitochondrial failure and cellular stress responses such as autophagy and apoptosis [20]. Moreover, mitochondrial functions extend beyond the boundaries of the cell, influencing the physiology of the organism by regulating communication between cells and tissues. Hence, focus has been shifted to mitochondrial dysfunction as one of the central factors in AD pathogenesis and development.

Mitochondria are the power generators of the cell, supplying most of the ATP through oxidative phosphorylation (OXPHOS). Neurons have among the highest rates of ATP consumption among all cell types, primarily to maintain the ionic gradients required for ongoing electrophysiological activity, neurotransmission, and short-term synaptic plasticity [21]. For instance, ATP-dependent processes are required to maintain the conditions required for efficient intracellular calcium (Ca^2+^) signaling, such as low basal cytoplasmic concentrations and storage in Ca^2+^ release organelles [22]. Moreover, damaged mitochondria are major sites of free radical generation and can directly trigger apoptosis via cytosolic release of cytochrome c (Cyto c) [23]. Thus, neurons may be damaged by even a modest decrease in mitochondrial function.

With age, mitochondria accumulate oxidative damage while reparative efficiency is reduced, resulting in deficient cellular bioenergetics (termed the “mitochondrial theory of aging”). In turn, metabolic disruption leads to reduced neuronal resistance to other sources of damage and ensuing neurodegeneration as well as impaired synaptic plasticity and associated cognitive deficits. Free radical generation by mitochondria is caused by the leakage of electrons during the electron transport cascade for OXPHOS, resulting in oxidative stress and disruption of cellular metabolic and signaling pathways. At the cellular level, bioenergetic insufficiency results in reduced Ca^2+^ buffering capacity, apoptosis, and a progressive decline in postmitotic cell number, potentially exacerbating age-associated disorders such as AD [24]. Recently, a mitochondrial cascade hypothesis was proposed emphasizing the role of mitochondrial bioenergetics in AD [24,25,26,27]. This hypothesis further states that Aβ is an epiphenomenon of AD pathology rather than the primary cause, a notion consistent with the finding that early mitochondrial dysfunction can lead to cognitive impairment, increased Aβ aggregation, and AD pathogenesis [25,26]. Neurons possess a higher mitochondrial count than most other cell types, especially at synapses, and it is plausible that a fall in viable number below a certain threshold triggers a cascade of events resulting in impaired neuronal function and intercellular signaling. Studies have highlighted the presence of damaged mitochondria near synapses resulting from various insults, including accrued Aβ oligomers or fibrils and phosphorylated tau. Conversely, dysfunction due to mitochondrial DNA (mtDNA) damage, mutations, or impaired transport of metabolites and proteins may lead to Aβ oligomeric or fibrillar formation and phosphorylated tau accumulation [28]. Thus, accrued mitochondrial damage due to insults such as oxidative stress and accumulation of Aβ oligomers may act synergistically to further damage mitochondria and drive AD pathology [27]. Therefore, mitochondrial quality control (mtQC) may be central to maintaining a healthy pool of neurons during aging, while a mutually reinforcing cycle of mitochondrial dysfunction and Aβ formation may lead to mitochondrial loss, metabolic failure, and neuronal death (Figure 1).

Several recent studies have highlighted the involvement of mitochondrial dysfunction in AD [26,27,28,29]. Besides, availability of information related to mitochondrial proteomic has opened the door that uncovers various aspects of this energy associated organelle, which further adds on the understanding of mitochondrial dysfunction in AD [28]. Thus, sustaining mitochondrial function therapeutically may forestall neurodegeneration and loss of cognition in AD [30,31]. Given the likely importance of mitochondrial dysfunction in AD pathogenesis, this work summarizes the molecular mechanisms for mitochondrial dysfunction contributing to cognitive impairment in AD. Additionally, we review the available drugs and techniques targeting mitochondria for AD treatment.

## 2. Factors Debilitating Mitochondrial Function

### 2.1. Aging

Advanced age is a major risk for multiple neurodegenerative disorders both due to accrued damage and impaired self-repair. Aging is associated with marked changes in mitochondria structure and function. For instance, several reports have revealed profound age-dependent changes in mitochondrial membrane architecture, such as the disappearance of cristae and inner membrane vesicles. Furthermore, the dissociation of ATP synthase dimers into monomers ruptures the outer membrane and releases apoptogens into the cytoplasm. In addition, inner membrane vesiculation and dissociation of ATP synthase dimers impairs the capacity to maintain sufficient ATP (energy charge) for essential cellular functions. Age-dependent accumulation of somatic mtDNA deletions and base substitutions results in reduced expression of mtDNA-encoded OXPHOS enzymes, enhancing superoxide (O_2_^−^) production at OXPHOS complexes I/III and reducing ATP generation efficiency, thereby increasing the probability of metabolic failure and apoptosis. Impaired mitochondrial homeostasis during aging appears to be associated with disequilibrium between fusion and fission, with the predominance of fission over fusion preventing functional complementation of damaged mitochondria and thus accelerating deterioration [32]. Further, age-dependent accumulation of synaptic mitochondria is reported to interfere with synaptic activities, including the ATP production, and calcium homeostasis required for efficient depolarization-evoked release of neurotransmitter vesicles and plasticity, thereby impairing cognition and memory [33,34]. Compared to nonsynaptic mitochondria, synaptic mitochondria are more prone to age-dependent alterations and accumulation of Aβ aggregates.

Aging is the most important risk factor for the development of sporadic AD, which increases in prevalence from only 2% between 65–69 years to 25% in individuals older than 90 [35]. Therefore, several cohort studies have emphasized that age must be considered when assessing the likely efficacy and safety of interventions against AD [2]. In addition to metabolic failure, the aging process may be driven by accumulation of free radical damage. As mitochondria are the principal source of free radicals in the cell, oxidative damage would be most severe to mitochondrial macromolecules, in particular mtDNA [36]. Prolonged accumulation of free radicals is further associated with reduced activities of antioxidant enzymes, such as superoxide dismutase (SOD), catalase (CAT), glutathione peroxidase (GPx), and glutathione reductase in the AD brain [37,38].

Furthermore, increased free radical generation, reduced ATP synthesis, loss of tissue regeneration capacity, and impaired DNA repair mechanisms will contribute to misfolded protein formation and aggregation, including aggregates that enhance the activities of the APP lytic enzymes β- and γ-secretase. This enhancement augments APP amyloidogenic processing, leading to Aβ plaque formation and resulting in AD-associated impairments, such as focal neurodegeneration and cognitive decline [36,39]. Further, a secondary mitochondrial cascade may mediate damage caused by a primary Aβ cascade, or Aβ produced as part of a primary mitochondrial cascade could itself cause harm [27]. Reports also suggest that an aging-mediated reduction in proteasome activity may further promote Aβ and tau accumulation [40,41]. Collectively, these aging-related processes form a vicious cycle, resulting in progressive increases in mitochondrial dysfunction as well as Aβ and tau accumulation, the two major pathological hallmarks of AD.

### 2.2. Genetically Induced Mitochondrial Dysfunction

Alzheimer’s disease is characterized by the abnormal processing and accumulation of mutant or damaged intracellular and extracellular proteins, increasing neuronal vulnerability to dysfunction. Several studies have revealed a higher occurrence of both nuclear and mtDNA mutations in AD brains [42,43]. Genetic mutations in presenilin 1 (PSEN1), presenilin 2 (PSEN2), APP, tau, and APOE4 genes are strongly associated with Aβ aggregation and AD development [28,44]. Among these genes, mutations in APP (21q21), PSEN1 (14q24), and PSEN2 (1q42) are fully penetrant and follow an autosomal dominant inheritance pattern, resulting in aggressive forms of early-onset AD that account for approximately 5% of all AD cases [45]. In addition, other susceptibility genes are believed to increase AD risk or drive pathogenesis through complex interactions with environmental factors. For instance, allele polymorphisms of APOE (19q13) are associated with increased risk for later-onset AD, typically at 65 years or older [46].

Mitochondria DNA is composed of complementary H- and L-strands in a circular form with an approximate length of 15.5 kb [25]. Human mtDNA possesses 37 genes, of which 13 encode for respiratory and electron transport chain (ETC) components, 2 for ribosomal (r)RNAs, and 22 for transfer (t)RNAs necessary for mitochondrial protein synthesis [47]. Among the 13 mtDNA genes associated with the respiratory system and ETC, seven encodes complex I (NADH dehydrogenase, ND) components ND1, ND2, ND3, ND4, ND4L, ND5, and ND6, one encodes the complex III (cytrochrome reductase), three encode complex IV (cyt c oxidase or COX) components COX I, COX II, and COX III, and two encode complex V (ATP synthase) components ATPase6 and ATPase8. Other subunits of complex II are encoded by nuclear nDNA. While mtDNA is critical for the proper functioning of mitochondria, it is prone to oxidative damage due to proximity with free radical generation sites, the lack of DNA-protective histones, and less efficient DNA repair mechanisms; indeed, it is estimated that mtDNA is ten times more prone to mutations than nDNA [48]. These mutations can occur at sites of known mtDNA transcription and replication regulatory elements, disrupting ETC component expression, and mitochondrial homeostasis. Moreover, mtDNA mutations (e.g., partial deletions, duplications, and point mutation) can be inherited maternally. These mutations then propagate through clonal expansion, eventually reaching a threshold for significant deleterious effects on mitochondrial function, resulting in cell death, and other age-related changes. Mitochondrial dysfunction is strongly associated with cognitive deficits and dementia due to high metabolic requirements, including for the synaptoplastic processes underlying higher order cognitive functions [49]. For instance, Tanaka et al. (2008) reported impaired retention and consolidation of memory traces in mice harboring mtDNA mutations (Mito-mice) [50]. Hence, mutations in mtDNA, such as somatic mutations in OXPHOS genes, as well as several allelic polymorphisms (e.g., in APOE) are strongly associated with particular pathophysiological features of AD, including cognitive impairment and OS from excessive free radical generation [51,52,53].

### 2.3. Environmental Toxins and Mitochondrial Dysfunction

Recent studies have proposed that various lifestyle × environmental interactions can increase AD risk. Mitochondrial dysfunction is exacerbated by toxins such as pesticides, organic pollutants, heavy metals (Pb, MeHg, Cd, and As), xenobiotic compounds, industrial toxic waste products, and poly-chlorinated biphenyls among others. Exposure to such toxic agents has been well documented to cause neurotoxicity via mitochondrial dysfunction, leading to symptoms similar to those of AD [54,55]. Much ongoing research aims to determine the direct role of metal toxicity in mitochondrial dysfunction and AD progression [56]. In addition to mitochondrial function and morphology, some toxins have been shown to impair mtQC systems in AD models. For example, high-dose scopolamine administration induced several AD critical features in experimental animals, such as free radical generation, cognitive impairment, and accumulation of Aβ [57,58], and to increase mitochondrial vulnerability to swelling and membrane potential (ΔΨm) dissipation, which is required for coupling of the ETC to ATP production [59]. Streptozotocin was also found to induce cognitive impairments and brain accumulation of both Aβ and hyperphosphorylated tau [60,61]. Further, treatment was associated with decreased activity of complex I and complex IV, increased dynamin-related protein 1 (Drp1) protein expression, and decreased (depolarized) ΔΨm [60,62]. Treatment of cells with Aβ has also been found to induce mitochondrial defects, including ΔΨm depolarization, membrane fragmentation, and generation of free radicals. These examples strongly suggest that mitochondrial dysfunction contributes to AD risk and pathogenesis, underscoring mitochondria as promising targets for AD treatment.

### 2.4. Metabolic Syndrome (MetS)-Induced Mitochondrial Dysfunction

Multiple lines of evidence indicate that an imbalanced diet (high-calorie or calorie-deficient) results in mitochondrial failure and metabolic syndrome (MetS), which in turn can increase the risk for AD [63]. Characteristic features of MetS include metabolic disturbances that facilitate tissue stress and dysfunction leading to insulin resistance (IR), a closely related event to mitochondrial dysfunction [64]. Hyperglycemia due to obesity is reported to disrupt the normal TCA cycle in the mitochondrial matrix [65]. Middle-age obesity is strongly correlated with cognitive impairment and increased susceptibility to AD [63,66]. Further, MetS also results in accumulation of advanced glycation end products (AGES) that can initiate hyperactive free radical generation, leading to oxidative stress, and seed aggregation of Aβ fibrils in neurons and microglia [67]. Mounting evidence suggests a complex reciprocal association between mitochondria dysfunction and MetS disorders such as diabetes, obesity, and non-alcoholic fatty liver disease, which may in turn elevated AD risk. Further study is required to verify associations between MetS and AD and to elucidate the underlying molecular pathways linking these disorders.

## 3. Current Etiological Hypothesis of AD Involving Mitochondrial Dysfunction

### 3.1. Deficits in Mitochondrial Bioenergetics

The brain consumes more energy per unit weight than any other tissue, accounting for 20% of total body energy expenditure at baseline. Hence, brain tissue is exquisitely sensitive to any disruption or reduction in ATP generation by OXPHOS. During OXPHOS, an electrochemical gradient is produced between the inner membrane and the mitochondria matrix that drives ATP synthesis (Figure 2). This process is mediated by five mitochondrial protein complexes (I–V). Complex I is a sophisticated microscale pump consisting of 45 core subunits and requiring a multitude of additional assembly factors for biogenesis [68]. It initiates ETC by removing two electrons from NADH, which are then moved sequentially through the remaining complexes. This process is couple to movement of protons from the mitochondrial matrix to the intermembrane space. The proton gradient generated by complexes I, III, and IV is then use to power the rotary turbine-like ATP synthase of complex V, which drives phosphorylation of ADP to ATP [69]. The energy stored in ATP is then used to meet the immediate energy requirements of cells as well as for biosynthesis, including the synthesis of neurotransmitters, and complex phosphorylation-mediated signaling networks.

Impaired mitochondrial OXPHOS efficiency, dysfunction of mitochondrial respiratory enzymes, and decreased mitochondrial complex (I–V) activities have been detected in the AD brain and various AD models [70]. For instance, complex IV deficiency has been measured in the platelets, fibroblasts, and postmortem brain tissue samples of patients with mild cognitive impairment (MCI), which often progresses to AD, as well as in diagnosed sporadic sAD patients. It has been proposed that the accumulation of mtDNA deletions with age may be responsible for this complex IV deficiency. Lindeboom and Weinstein (2004) and Reed et al. (2008) reported a substantial (~35%) decrease in ATP synthase activity among MCI patients [71,72]. Further, the cognitive deficits and synaptic plasticity impairments observed in AD were associated with a decrease in extracellular ATP [73]. Beck et al. (2016) also reported a 67% reduction in synaptic mitochondrial Fo-ATP synthase activity in an AD mouse model compared to age-matched controls, accompanied by a reduction in ATP, increased oxidative stress, and reduced ΔΨm due to opening of the mitochondrial permeability transition pore (mPTP), which results in release of apoptotic effectors into the cytoplasm [74].

Furthermore, low ATP levels are reported to correlate with impairments in long-term potentiation (LTP) and long-term depression (LTD), forms of associative synaptic plasticity at glutamatergic synapses strongly implicated in various forms of learning and memory [75]. A reduction in ΔΨm has also been observed in M17 neuroblastoma cells and transgenic mice overexpressing APP. Fibroblasts obtained from patients with PSEN1 mutations were found to show reduced basal and maximal respiration. Further, impaired respiratory capacity and reductions in basal oxygen consumption and spare capacity have been reported in PSEN2 knockout mouse embryonic fibroblasts, and these deficits were restored by overexpressing exogenous PSEN2 on the knockout background. While these findings suggest that metabolic insufficiency due to mitochondrial complex dysfunction contributes to brain pathology in AD, an increase in complex I activity has been observed in sAD platelets [76]. Further, the contribution of complex IV dysfunction to ATP deficiency in AD is still controversial.

These flaws in the OXPHOS machinery impede sufficient ATP production and enhance free radical generation, damaging mitochondrial proteins, activating the mPTP, and mutagenizing mtDNA, further contributing to defective OXPHOS. In addition to ATP production, the inner membrane electrochemical potential generated by OXPHOS is vital for mitochondrial protein import and for regulating molecular changes that alter mitochondrial responses to other insults, potentially leading to AD [77]. These mitochondrial defects ultimately contribute to the characteristic synapse loss in the neocortex and hippocampus of AD patients, which in turn is associated with cognitive impairments such as memory loss [78,79].

### 3.2. Deficits in Mitochondrial Dynamics

Mitochondria are dynamic organelles that constantly split (fission) and combine (fusion) to maintain cellular metabolic homeostasis. Mitochondrial fusion allows for the exchange of mitochondrial contents, such as membrane lipids and mtDNA, to reduce the proportion of damaged constituents. Fusion is orchestrated by three large GTPases, mitofusin 1 (Mfn1), mitofusin 2 (Mfn2), and optic atrophy protein 1 (OPA-1), which tether two neighboring mitochondria together, thereby facilitating the merging of the inner membranes (IMs) and outer membranes (OMs). To accomplish this function, the Mfn transmembrane proteins forms homo- or heterodimers spanning the OM and utilize GTPase activity to power fusion, while IM fusion is accomplished by the interaction of OPA-1 facing the intermembrane space with an Mfn. In contrast, mitochondrial fission can promote both increasing the number of organelle and the degradation, sequestration, and elimination of irreversibly damaged mitochondria and constituents. Mitochondrial fission also involves a large GTPase, Drp1 as well as mitochondrial fission 1 (Fis1) localized to the OM and several other fission proteins. When a mitochondrion signals to divide, Drp1 translocates to the OM and initiates the process by oligomerizing into large ring-like complexes circling the future fission site to physically pinch the organelle into two daughter mitochondria. The balance between mitochondrial fission and fusion is susceptible to physiological and pathophysiological conditions within the cell, and imbalance can lead to reversible mitochondrial swelling, fragmentation, depolarization, and morphological alterations, which in turn may increase susceptibility to other forms of neuronal stress such as NMDA receptor (NMDAR)-mediated glutamate toxicity (excitotoxicity) and ultimately to cell death [80].

In fact, mitochondrial homeostasis is disrupted in both sporadic and inherited forms of AD. Ultrastructural morphometric studies conducted using biopsied brain tissues from AD patients and transgenic mouse models have revealed significant structural damage to mitochondria, including broken cristae and even near total loss of the inner structure in vulnerable pyramidal neurons. Such conditions likely arise from mitochondrial dysfunction and elevated free radical generation. A slight but significant increase in mitochondrial size and reduction in number were observed in AD neurons, suggesting the involvement of abnormal mitochondrial dynamics in pathogenesis. A similar reduced mitochondrial count has also been reported in PSEN1 mutant fibroblasts. Further, shorten mitochondria were observed in fibroblasts from sAD patients, while another study reported an increase in the number of fragmented mitochondria [30].

However, these abnormal changes in mitochondrial morphology do not necessarily indicate increased mitochondrial fusion as similarly looking swollen mitochondria were observed in fusion-deficient Mfn2 mouse brain. However, the elongated mitochondria observed in AD fibroblasts are distinct from those of age-matched normal human fibroblasts, which are predominantly sausage-shaped. Reports also suggest that these morphological differences may be due to differential expression levels of fission proteins (Drp1 and Fis1) and fusion proteins (Mfn1, Mfn2, and OPA1) as observed in AD fibroblasts and AD brain. For instance, a significant reduction in Drp1 and concomitant increase in Fis1 expression have been reported in peripheral blood lymphocytes obtained from AD and MCI patients, indicating that these deficits in mitochondrial dynamics are systemic. Further, peripheral blood cells from AD patients also demonstrated mitochondrial dysfunction and oxidative stress. Excessive mitochondrial fission has been unequivocally demonstrated in cells overexpressing APP, together with elevated Aβ production, mitochondrial fragmentation, heightened ROS production, reduced ATP generation, lower ΔΨm, and an increase in the number of mitochondria with structural damaged. Such mitochondrial deficits were even more severe in M17 cells expressing the familial AD-causing Swedish APP mutation. Similarly, primary cortical neurons treated with okadaic acid also demonstrated mitochondrial fragmentation accompanied by decreased total Drp1, increased phosphorylated Drp1, and elevated ROS production.

### 3.3. Mitochondrial Transport

Mitochondria are not static organelles but demonstrate high motility powered by various GTPases. This intracellular movement is essential for maintaining normal neuronal polarity, membrane potential, neurotransmission, and synaptic plasticity. Mitochondria are particularly abundant in synapses, and obstruction of mitochondrial movement can markedly disrupt synaptic communication among neurons [81]. Kinesin is responsible for moving mitochondria in the anterograde direction toward the nerve terminal, while dynein moves mitochondria in the retrograde direction toward the soma (Figure 3). While AD etiology remains enigmatic, it is established that axonal pathology and synaptic dysfunction occur before detectable Aβ and tau aggregation and that mitochondrial dysfunction results in synaptic failure [82]. Axonal degeneration due to abnormal accumulation of mitochondria has been observed within large swellings along dystrophic and degenerating neurites in the AD brain [82]. Therefore, preventing mitochondrial dysfunction appears to be a viable strategy to maintain synaptic strength and plasticity, thereby delaying cognitive decline in AD patients. Beck et al. (2016) found that loss of oligomycin-sensitivity conferring protein (OSCP), a component of the mitochondrial ATP synthase, induces synaptic dysfunction [83], while restoration of OSCP mitigated Aβ-mediated mitochondrial dysfunction and further preserved synaptic function as evidenced by sustained density and transmission.

Further, impaired anterograde movement has been detected in an APP mouse model, while both retrograde and anterograde transport were impaired in PSEN1 and APP/PSEN1 mouse models [84]. The authors also noted that neurons with impaired mitochondrial transport were more susceptible to excitotoxic cell death. Calkins and Reddy (2011) reported a notable decrease in mitochondrial movement within Aβ-treated mouse hippocampal neurons that was associated with abnormal morphology due to defects in fusion and fission [85]. Abnormal mitochondrial distribution and round, swollen, and damaged mitochondria were also documented, coinciding with the loss of mitochondrial oxidative activity but preceding the onset of amyloid plaque formation and memory impairment in Tg2576 APP transgenic mice and APP/PS1 mice. Collapse of the mitochondrial network and transportation is also a critical factor in synapse loss associated with the cognitive decline in AD [86]. However, the exact mechanisms underlying this interplay among APP processing, mitochondrial motility, and synaptic loss remain to be determined. An ultrastructural study reported loss of the normal microtubular structure around intracellular Aβ peptide, which could impair mitochondria movement and initiate apoptotic cascades in synapses and dendrites, resulting in early synaptic degeneration, a central characteristic of AD [87]. Several reports have suggested a related pathomechanism involving Ca^2+^ dysregulation and changes to microtubule-associated proteins such as tau, which acts as part of the “rail track” for the rapid delivery of mitochondria to sites of high energy demand. Increased local Ca^2+^ has been suggested to arrest this mitochondrial transport, resulting in local energy deficits. Further, Ca^2+^ elevation may also trigger overexpression and hyperphosphorylation of tau via activation of kinases or microsomal prostaglandin E synthase 1 (mPGES1) [88], thereby disrupting mitochondrial distribution and leading to axonal dysfunction and synapse loss as observed both in vitro and in AD model mice [89,90]. Increasing evidence also demonstrates that, in addition to aberrant phosphorylation, caspase cleavage of tau plays a critical role in the oligomerization and formation of pathological tau species in AD.

### 3.4. Mitochondrial Quality Control (mtQC)-Associated Deficits

In healthy cells, damaged mitochondria are eliminated by autophagy, a controlled degradative process that recycles the remnants for metabolism and biosynthesis. Under pathological conditions, autophagy is a critical survival mechanism and pathway for mitochondrial quality control (mtQC), eliminating those organelles producing excessive free radicals and disrupting other cellular functions [91,92]. The activation of specific mtQC mechanisms is dependent on the extent of mitochondrial damage at both molecular and organelle levels, such that mtQC mechanisms form an interdependent hierarchical system that monitors mitochondrial integrity, thereby ensuring cell survival [93]. Recent studies have demonstrated that mitochondria act as “guardians in the cytosol” by importing and degrading aggregation-prone cytosolic proteins via specific proteases and chaperones. This mitochondrial proteostasis may also act as the first line of defense against mitochondrial protein damage [94]. For instance, mitochondrial chaperones such as heat shock proteins (HSPs) 22, 60, and 70 help refold proteins to their native three-dimensional conformation, thereby maintaining functionality [95,96]. In contrast, irreversibly damaged proteins are degraded by a set of mitochondrial resident proteases, primarily Lon and Clp proteases [97,98]. These ATP-dependent proteases recognize the exposed hydrophobic regions of denatured proteins and degrade them after unfolding [99]. Defects in these proteases impair the capability of mitochondria to monitor, repair, and remove damaged proteins, which eventually induces misfolded protein aggregation within mitochondria and causes mitochondrial dysfunction. Furthermore, mutations in genes encoding these proteases and chaperones are also associated with several neurological disease phenotypes [100,101]. Elevation of misfolded and damaged proteins rises above a certain threshold activates the mitochondrial unfolded protein response (mtUPR), resulting in elevated expression of nuclear genes encoding mitochondrial chaperones and proteases, thereby reducing the concentration of damaged proteins [102,103]. Recently, upregulation of these mitochondrial proteases and chaperones has been detected in AD patients and 3XTgAD preceding amyloid and tau pathology, suggesting that failure of mitochondrial proteostasis concomitant with general mitochondrial dysfunction is an early event in AD progression [83,104].

In addition to the mtUPR, damaged mitochondrial proteins can also be degraded with the help of the cytoplasmic 26S proteasome system. Here, damaged OM proteins are retro-translocated from the membrane with the help of the AAA+ ATPase p97, and then degraded by the cytoplasmic proteasome system [105]. The next level of mtQC becomes activated when there is localized or severe damage. The two mechanisms that function in response to these forms of damage, constituting the third and fourth levels of mtQC, are mitochondrial-derived vesicles (MDVs) and a mitochondria-specific autophagic pathway termed mitophagy. The MDVs are 70–150-nm vesicles that bud off from the mitochondria, selectively incorporating large assemblies of damaged proteins and lipids from the OM, IM, and matrix [106]. They are usually formed when there is local accumulation of damaged proteins resulting in blockage of mitochondrial import channels. These MDVs containing damaged mitochondrial proteins are subsequently targeted to lysosomes, late endosomes, multivesicular bodies, and peroxisomes, or undergo exocytosis [106]. However, the contribution of MDVs to mtQC and relevance to AD are relatively underexplored.

Mitophagy differs from the other aforementioned pathways in that it can target an entire damaged mitochondrion for sequestration and degradation via the mitophagosome–lysosome pathway [107]. Accumulation of damaged mitochondria due to a reduction in mitophagic degradation was observed in the soma of vulnerable AD neurons. Distinct mitophagy pathways become activated by specific cues, and a plethora of proteins are involved in the execution of these pathways. Reports suggest that the parkin pathway is robustly induced upon mitochondrial damage in AD patient brains and animal models of AD [108]. This pathway involves stabilization/activation of PINK1 at the OM by impaired ΔΨm [109,110]. In turn, PINK1 recruits and phosphorylates the E3-ubiquitin ligase parkin to drive ubiquitination of mitochondrial OM proteins, which labels damaged mitochondria for degradation through the mitophagy pathway.

Cytosolic parkin was found to be progressively depleted in brain during AD progression, resulting in mitophagic pathology, and augmented mitochondrial defects. Another study reported diminished parkin as well as abnormal PINK1 accumulation in AD patient-derived skin fibroblasts and brain biopsy tissue. Fibroblasts and iPSC-derived neurons from AD patients also exhibited mitochondrial localization of parkin, suggesting that mitochondria were labeled correctly for mitophagy but not degraded. Consistent with this notion, restoration of mitophagy by overexpression of parkin, as evidenced by decreased PINK1, rescued ΔΨm and reduced retention of defective mitochondria. Collectively, these findings suggest that impaired mitochondrial function and abnormal retention of dysfunctional mitochondria in neurons of AD patients may stem from mitophagy defects.

Defects in activation of the autophagy/mitophagy initiator proteins ULK1 and TBK1 were observed in postmortem hippocampal tissues from AD patients, cortical neurons derived from patient induced pluripotent stem cell (iPSC), as well as in the brains of AD mouse models. Moreover, pathogenic truncation of tau may also impair mitophagy leading to improper mitochondrial turnover. The degradation capacity of lysosomes is also critical for mitophagic clearance, and defects in lysosomal proteolysis of autophagic cargoes have been reported in AD brains and linked to AD pathogenesis. Further, suppression of lysosomal proteolysis in wild-type mice mimicked AD neuropathology and exacerbated autophagic pathology and amyloidogenesis. In addition, mutations in PSEN1/ApoE4 also disrupted lysosomal function, further supporting the view that defective mitophagy is likely an early event in AD and plays a causative role in AD-linked neuropathology. Other factors, including Aβ peptides, phospho-tau, ROS, and oxidized lipids and lipoproteins, could also impair lysosomal proteolysis, resulting in toxic protein accumulation, triggering apoptosis, and neuronal death in AD.

## 4. Crosstalk between Mitochondrial Dysfunction and Calcium Signaling

Healthy mitochondria act both as a high-capacity calcium buffer and as a source of energy for transmembrane Ca^2+^ pumps and exchangers in the plasma membrane and endoplasmic reticulum (ER), so mitochondrial dysfunction markedly impairs calcium homeostasis [111]. In the normal physiological state, increases in cytosolic Ca^2+^ concentration are sequestered through IM and OM channels to sustain normal Ca^2+^ signaling and prevent cytotoxicity (Figure 4) [112,113]. Insufficient buffering disrupts muscle contraction, synaptic communication, neurotransmitter release, and signal transduction, and contributes to the neuronal loss underlying memory impairment in AD [33]. Further, Ca^2+^ signaling is vital for the changes in gene expression, dendritic spine morphology, and spinogenesis underlying long-term synaptic plasticity [114]. Reports suggest that mitochondrial dysfunction and Ca^2+^ dysregulation are upstream events for Aβ aggregation and contribute to motor coordination deficits in PS1-FAD patients, PS1-FAD murine model, and 3xTg-AD model mice prior to Aβ aggregation [115,116]. For these reasons, elevated intracellular Ca^2+^ due to insufficient mitochondrial buffering is regarded as a core mechanism linking amyloid accumulation to neuronal cell death and cognitive decline [116].

Both PSEN1 and PSEN2 are reported to be localized at mitochondria-associated membranes (MAM) and to play a key role in Ca^2+^ homeostasis [117,118]. PSEN2 modulates Ca^2+^ uptake into ER and mitochondria, and its overexpression was observed to increase the interaction between organelles, leading to the increased mitochondrial Ca^2+^ uptake, activation of the mPTP, activation of calpains, and generation of free radicals underlying learning and memory impairments in fAD-AD models [118,119].

Excessive cytosolic Ca^2+^ also causes tau hyperphosphorylation, which induces misfolding and detachment from microtubules, resulting in its aggregation and translocation to the somatodendritic compartment where it forms neurofibrillary tangles, a pathological hallmark of AD. This condition in turn results in the disruption of mitochondrial transport, energy deprivation, oxidative stress at the synapse, and eventually in neurodegeneration [88].

## 5. Free Radical Generation and Mitochondrial Dysfunction

Free radicals such as superoxide (O_2_−) and hydroxyl radicals (OH) as well as non-radicals that can form free radicals under cellular conditions such as hydrogen peroxide (H_2_O_2_) are produced during mitochondrial electron transport and other reactions. These substances are characterized by high reactivity to biomolecules such as amino acids, lipids, and nucleic acids. Due to their short lifespan, free radicals are most likely to react with proximal biomolecules, such as mtDNA and proteins. Superoxide radicals are formed primarily by complexes I, complex III, and TCA components, such as α-ketoglutarate dehydrogenase, and are released into the IMM space and matrix. These mitochondrially generated free radicals can also enter the cytoplasm via voltage-dependent anion channels to induce the oxidation of membrane lipids and proteins.

Excessive free radical generation caused by mitochondrial damage and/or a reduction in endogenous antioxidant capacity will result in deleterious chemical alterations to biomolecules, termed oxidative stress. Signs of oxidative stress are found among neurons both in the AD brain and various animal models and are associated with energy deficiency due to reduced ATP synthase activity. In addition, free radicals are reported to promote the expression and activity of β- and γ-secretases, leading to increased Aβ production from APP, consequently exacerbating mitochondrial dysfunction [86,120]. Free radical elevation and ensuing oxidative damage are more pervasive than Aβ plaques and tau tangles, suggesting greater importance in early pathogenesis [121]. Oxidative stress also reduces the expression of genes encoding mitochondrial ETC subunits, resulting in lower ATP production. This may in turn impair Ca^2+^ homeostasis, which elevates free radical generation from various mitochondrial and cytosolic reactions, and ultimately may activate the mPTP, allowing the translocation of pro-apoptotic molecules from the mitochondria to cytosol and promoting neuronal cell death and other downstream stress-activated pathways such as neuroinflammation. Thus, oxidative stress from mitochondrial dysfunction can create a self-sustaining cycle that further exacerbates mitochondrial dysfunction, calcium dysregulation, oxidative stress, and Aβ formation, ultimately accelerating neuronal dysfunction, neurodegeneration, and cognitive impairment [121,122]. Indeed, the degree of cognitive impairment and synaptic loss in AD has been related to the amount of Aβ accumulated in mitochondria [122].

## 6. Therapeutic Interventions

A greater understanding of the mechanisms contributing to mitochondrial impairment, neurodegeneration, and cognitive dysfunction as summarized here may soon lead to more effective therapeutic techniques. While not a cure, therapies targeting mitochondrial dysfunction may slow or halt the progression of AD. Table 1 summarizes therapeutic agents and strategies targeting mitochondria that have been studied for improvement of cognition and memory defects in AD models. These include mitochondrially targeted antioxidants as well as modulators of mitochondrial dynamics and mQC. A major focus of these studies is reduction of free radical generation or damage, microglial activation, and excessive mitochondrial fragmentation, thereby minimizing mitochondrial dysfunction, neuronal injury, and cognitive deficits during AD progression.

## 7. Concluding Remarks

In this review, we argue that mitochondrial dysfunction contributes to the pathogenesis of AD through complex interactions with other pathomechanisms. Exposure to environmental toxins, metabolic disorders, impaired OXPHOS, and mtDNA mutation are all possible causes for mitochondrial dysfunction, which in turn leads to reduced ATP production, greater ROS degeneration, altered mitophagy and ensuing accumulation of defective organelles, reduced motility, imbalanced fission and fusion, impaired protein and metabolite transport, and reduced calcium buffering. These deficits in turn trigger or exacerbate energy insufficiency, oxidative stress, calcium deregulation, misfolded protein aggregation, and excitotoxicity, ultimately leading to mitochondrial membrane potential dissipation, translocation of cytochrome c, neuronal apoptosis, and cognitive dysfunction. Further, these processes are interactive and mutually reinforcing, driving AD progression. However, these pathological changes are amenable to perturbation by certain treatments, raising hope that mitochondria-targeted interventions will one day slow or halt AD progression. We hope that the insights provided by this review may soon translate into practical and novel treatments for mitochondrial dysfunction in AD.

## Figures and Tables

**Figure 1 ijms-22-04850-f001:**
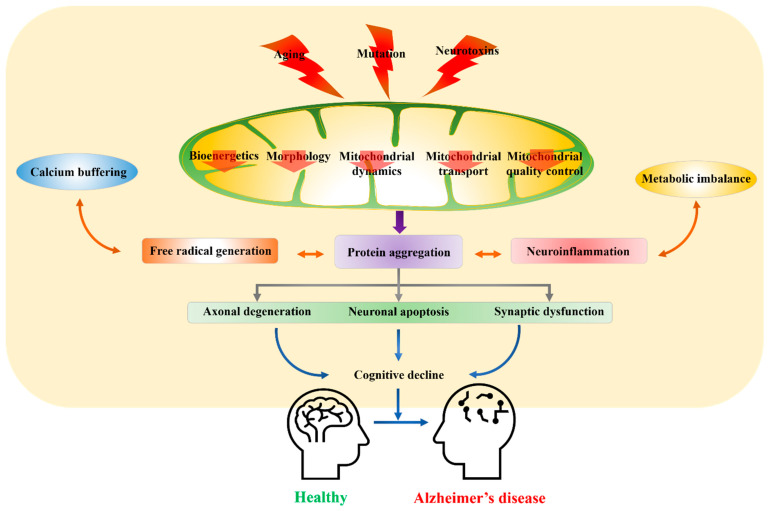
Mitochondrial dysfunction as a driver of cognitive impairment in Alzheimer’s disease (AD). Mitochondria are susceptible to various age-associated processes, mutations, and toxic insults such as metal exposure. Damaged mtDNA and other macromolecules accumulate during aging, leading to metabolic impairment. Resultant accumulation of damaged and dysfunctional mitochondria has been reported as an early sign preceding AD and contributing to disease progression. Dysfunctional mitochondria further cause bioenergetic deficiency, intracellular calcium dysregulation, and generation of free radicals leading to oxidative stress, thereby aggravating the effect of Aβ and tau pathology and further exacerbating mitochondrial damage, synaptic dysfunction, cognitive impairment, and memory loss.

**Figure 2 ijms-22-04850-f002:**
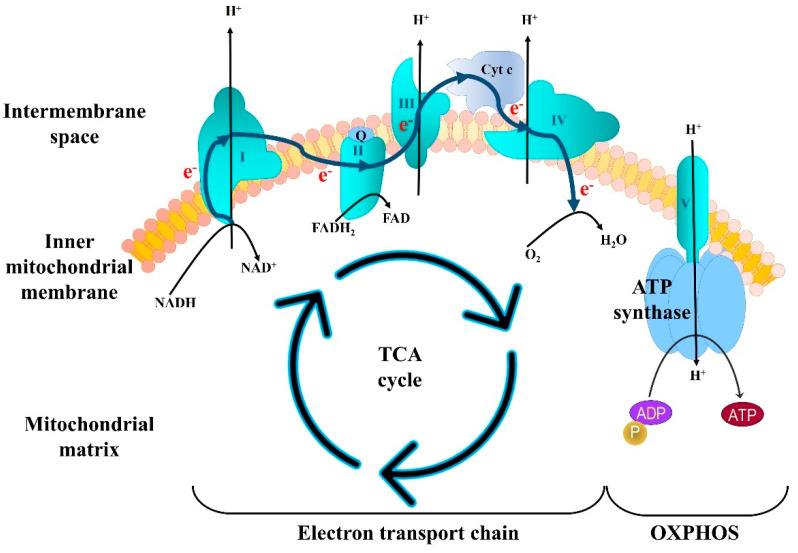
Mitochondrial energy metabolism. In healthy cells, the tricarboxylic acid (TCA) cycle works to reduce nicotinamide adenine dinucleotide (NADH) and oxidize succinate molecules, which are further used by the electron transport chain (ETC) to generate an electrochemical gradient between the inner membrane space and matrix. Mitochondrial Complex V (ATP synthase) uses this gradient to produce ATP.

**Figure 3 ijms-22-04850-f003:**
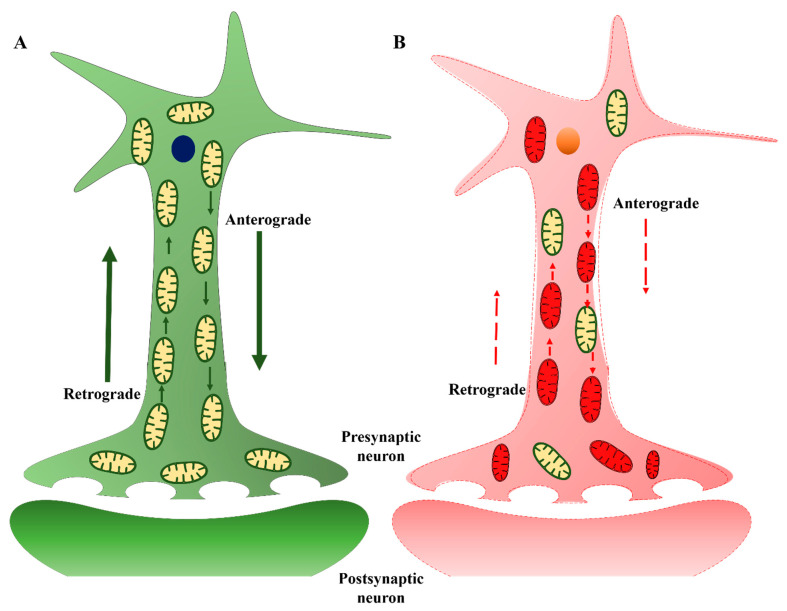
Neuronal mitochondrial trafficking is largely interrupted in AD. (**A**). In healthy neurons, mitochondria move from the cell body to axons, dendrites, and synapses by an anterograde mechanism, supplying ATP to nerve terminals. Mitochondria then travel back to the cell body from synapses through a retrograde mechanism. (**B**). In AD neurons, these mechanisms are disrupted primarily due to defective or functionally inactive mitochondria.

**Figure 4 ijms-22-04850-f004:**
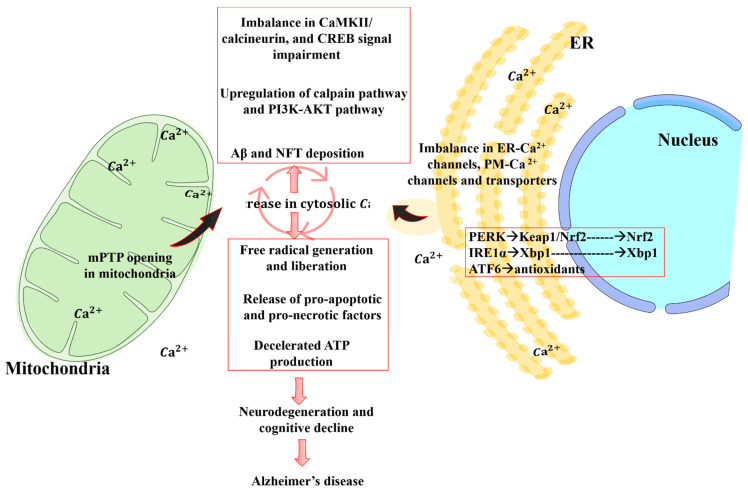
Crosstalk between mitochondria and endoplasmic reticulum (ER) during AD. Mitochondria and ER are interconnected via a specialized set of proteins, thus forming specific microdomains called mitochondria-associated ER membranes (MAMs). MAMs play an important role in calcium and lipid homeostasis, mitochondrial dynamics, and autophagy. Perturbations in ER–mitochondria interactions are implicated in AD progression, including neuronal cell death.

**Table 1 ijms-22-04850-t001:** Main therapeutic agents and strategies targeting mitochondria that have been explored to improve cognition and memory defects in AD.

Therapeutic Agent	Mode of Action	Mechanistic Pathways and Effects	References
**Mitochondrial Targeted Synthetic Analogs/Peptides**			
MitoQ	Enhances electron transport chain (ETC) activity	Free radical scavenger, prevents mPTP opening (anti-apoptotic), enhances CREB signaling, and improves mitochondrial health	[123,124,125,126,127,128]
MitoVitE	Inhibits lipid peroxidation	Free radical scavenger, prevents apoptosis by inhibiting mPTP opening, cytochrome c release, and caspase-3 activity	[125,129,130]
MitoPBN	Inhibits lipid peroxidation	Anti-apoptotic	[131]
MitoTEMPO	Inhibits lipid peroxidation	Eliminates mitochondrial superoxide, inhibits lipid peroxidation, and maintains mtDNA fidelity and copy number	[132]
Idebenone	Enhances ETC activity	Free radical scavenger, protects mitochondrial complex and ETC	[133,134,135,136,137]
SS-31	Inhibits lipid peroxidation. Activates PGC1α	Free radical scavenger, maintains mitochondrial transport, prevents mitochondrial depolarization by enhancing mitochondrial processing peptidase expression, and prevents apoptosis by inhibiting cytochrome c release	[138,139,140,141]
Mdivi-1	Drp1 inhibitor	Decreases mitochondrial fission, reduces ROS generation, and enhances mitochondrial biogenesis	[142,143]
DDQ	Inhibits Aβ and Drp1 binding	Decreases fission, increases fusion, and increases PGC1α, Nrfl, Nrf2, and TFAM	[144]
P110	Drp1 inhibitor	Decreases fission and ROS, enhances MMP, and prevents apoptosis	[145]
Dynasore	Drp1 and mTORC1 inhibitor	Inhibits mitochondrial fission and enhances biogenesis and mitophagy	[146,147]
TEMPOL	Superoxide dismutase mimetic	Protects against MMP depolarization	[148]
EUK-134	Superoxide dismutase mimetic	Protects against MMP depolarization	[148]
**Naturally Present**			
Co-Q10	Enhances ETC activity	Mitigates free radicals and enhances mitochondrial biogenesis	[149,150,151,152,153]
Creatine	Maintains energy reserve capacity	Mitigates ROS and enhances ATP reserves	[154]
Vitamin E	Antioxidant	Free radical scavenger, decreases abnormal protein nitration	[155]
Vitamin C	Antioxidant	Free radical scavenger	[156]
Glutathione	Targets glutathione to the mitochondrion	Free radical scavenger and anti-apoptotic	[130]
**Natural Compounds and Their Synthetic Derivatives**			
Curcumin	Antioxidant	Mitophagy modulator,	[157,158]
Resveratrol	SIRT1 activator	regulates mitochondrial biogenesis, acts as an antioxidant, and anti-inflammatory	[159,160,161]
Sulforaphane	Nrf2 activator	Reduces ROS and maintains redox homeostasis, upregulates cytoprotective genes, and reduces inflammation	[162,163]
Bezafibrate	PGC1α activator	Increases mitochondrial biogenesis and ATP production	[164]
NAD+ precursors	Enhances NAD+ signaling	Enhances mitophagy and increases ROS resistance	[165,166]
DNP	Activates CREB and PGClα	Mitigates ROS, stimulates autophagy	[167]
Rapamycin	mTOR inhibitor	Enhances mitophagy	[168]
N-acetyl-L-cysteine	Precursor of GSH	Free radical scavenger, protects against mPTP opening, anti-apoptotic	[125]
**Nanoparticles**			
Triphenylphosphonium conjugated-Ceria (CeO_2_) nanoparticles	Reduces superoxides	Protects mitochondrial morphology from oxidative damage	[169]

## Data Availability

Not applicable.

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
