# Peer review of "Mitochondrial Dysfunction as a Driver of Cognitive Impairment in Alzheimer’s Disease"

_ijms, 2021, doi:10.3390/ijms22094850_

Round 1

Reviewer 1 Report

Dear authors! Thank you for such an in-depth review of the role of mitochondria in AD. There are some points I want to address:

1) Perhaps, the authors could also mention Abyadeh et al. (2021) published in Exp Rev Proteomics (PMID: 33874826), where proteomic approaches to detect mitochondrial dysfunction early in the AD is discussed.

2) A newer reference for Swerdlow is available (Journal of Alzheimer’s Disease 62 (2018) 1403–1416 DOI 10.3233/JAD-170585).

3) In Figure 3, I suggest that you make the text "Increase in cytosolic Ca2+" visible, at the moment it is hidden behind the other parts of the image.

Author Response

Reviewer 1

Dear authors! Thank you for such an in-depth review of the role of mitochondria in AD. There are some points I want to address:

1) Perhaps, the authors could also mention Abyadeh et al. (2021) published in Exp Rev Proteomics (PMID: 33874826), where proteomic approaches to detect mitochondrial dysfunction early in the AD is discussed.

Response: With the reviewer's suggestion, we have incorporated the newly information from Abyadeh et al. (2021) in the main text and cited it in line; 92, 99-101, 166.

2) A newer reference for Swerdlow is available (Journal of Alzheimer’s Disease 62 (2018) 1403–1416 DOI 10.3233/JAD-170585).

Response: As suggested, we have acquired the information from Swerdlow et al. (2008) (line 153-156) and the citation has been added to the reference list [Reference 27].

3) In Figure 3, I suggest that you make the text "Increase in cytosolic Ca2+" visible, at the moment it is hidden behind the other parts of the image.

Response: Many thanks for the suggestion. As suggested, we have modified the figure and reinserted it at the appropriate place.

Reviewer 2 Report

This is an extensive and detailed review of the interactions between mitochondrial function and Alzheimer’s disease. The manuscript is acceptable for publication, though the authors should consider some minor edits. Table 1 needs to be formatted. It is very difficult to sort the information in Column 3 by its row. The sentence “Recently, a mitochondrial cascade hypothesis was 80 proposed emphasizing the role of mitochondrial bioenergetics in AD” in Line 80-81 should include a reference. The paper could use a figure of mitochondrial energy metabolism. The authors describe this process and the enzymes in great detail. A figure would support several sections of this document and would provide a helpful visual reference.

Author Response

Reviewer 2
This is an extensive and detailed review of the interactions between mitochondrial function and Alzheimer’s disease. The manuscript is acceptable for publication, though the authors should consider some minor edits.

1) Table 1 needs to be formatted. It is very difficult to sort the information in Column 3 by its row.

Response: Many thanks for your suggestion. We have added borders and increase the font size for clear readership quality.

2) The sentence “Recently, a mitochondrial cascade hypothesis was 80 proposed emphasizing the role of mitochondrial bioenergetics in AD” in Line 80-81 should include a reference.

Response: We thank the reviewer for pointing this. We have now added the appropriate references (Line 81).

3) The paper could use a figure of mitochondrial energy metabolism. The authors describe this process and the enzymes in great detail. A figure would support several sections of this document and would provide a helpful visual reference.

Response: We appreciate the reviewer’s comment. As suggested, we have added Figure 2 to define mitochondrial energy metabolism.